Diosgenin biosynthesis pathway and its regulation in Dioscorea cirrhosa L.

Lin Yan 1
Hu Qiuyan 1
Ye Qiang 2
Zhang Haohua 2
Bao Ziyu 2
Li Yongping yplee614@163.com 3
Mo Luo Jian 52116124@qq.com 1
1 Dongguan Institute of Forestry Science , Dongguan , Guangdong , China
2 College of Forestry and Landscape Architecture, South China Agricultural University , Guangzhou , Guangdong , China
3 Key Laboratory for Quality Regulation of Tropical Horticultural Crops of Hainan Province, School of Horticulture, Hainan University , Haikou , Hainan , China
Cao Yunpeng
Electronic publication date: 2024 Jan 23
Publication date: 2024
Volume: 12
Electronic Location ID: e16702
Received 2023 Aug 9; Accepted 2023 Nov 29
Copyright: ©2024 Lin et al.
Copyright year: 2024
Copyright holder: Lin et al.
License: This is an open access article distributed under the terms of the Creative Commons Attribution License, which permits unrestricted use, distribution, reproduction and adaptation in any medium and for any purpose provided that it is properly attributed. For attribution, the original author(s), title, publication source (PeerJ) and either DOI or URL of the article must be cited.
License URL: https://creativecommons.org/licenses/by/4.0/

Keywords: Dioscorea cirrhosa L., Diosgenin, Transcriptome, CYP450 gene family

Funding: The authors received no funding for this work.

==============================
Dioscorea cirrhosa L. (D. cirrhosa) tuber is a traditional medicinal plant that is abundant in various pharmacological substances. Although diosgenin is commonly found in many Dioscoreaceae plants, its presence in D. cirrhosa remained uncertain. To address this, HPLC-MS/MS analysis was conducted and 13 diosgenin metabolites were identified in D. cirrhosa tuber. Furthermore, we utilized transcriptome data to identify 21 key enzymes and 43 unigenes that are involved in diosgenin biosynthesis, leading to a proposed pathway for diosgenin biosynthesis in D. cirrhosa. A total of 3,365 unigenes belonging to 82 transcription factor (TF) families were annotated, including MYB, AP2/ERF, bZIP, bHLH, WRKY, NAC, C2H2, C3H, SNF2 and Aux/IAA. Correlation analysis revealed that 22 TFs are strongly associated with diosgenin biosynthesis genes (—r2— > 0.9, P < 0.05). Moreover, our analysis of the CYP450 gene family identified 206 CYP450 genes (CYP450s), with 40 being potential CYP450s. Gene phylogenetic analysis revealed that these CYP450s were associated with sterol C-22 hydroxylase, sterol-14-demethylase and amyrin oxidase in diosgenin biosynthesis. Our findings lay a foundation for future genetic engineering studies aimed at improving the biosynthesis of diosgenin compounds in plants.

Introduction

The biosynthesis of triterpene diosgenin involves a downstream pathway that is regulated by cytochrome P450 monooxygenase (CYP450s) and uridine diphosphate (UDP)-dependent glycosyltransferases (UGTs), which control a series of cholesterol oxidation, hydroxylation and glycosylation reactions (Seki, Tamura & Muranaka, 2015). CYP450 and UGT genes in plants exhibit high diversity, contributing significantly to the structural variety of triterpene. The biosynthesis of sterols relies on CYP450s. For instance, CYP51G and CYP710A encode for obtusifoliol 14- α demethylase and sterol-22-desaturase, respectively (Bak et al., 1997; Morikawa, Mizutani & Ohta, 2006). The CYP97 family plays role in lutein biosynthesis. Arabidopsis CYP97A3 and CYP97C1 catalyze the hydroxylation of carotenoid β- and ɛ-rings in the lutein biosynthetic pathway, respectively (Kim & DellaPenna, 2006; Tian et al., 2004). The CYP73A subfamily has only one number, cinnamic acid 4-hydroxylase (C4H), which catalyzes cinnamic acid to form precursors of lignin and many other phenolic compounds. CYP98A catalyzes the 3′-hydroxylase of coumaric acid (Babu, Rao & Reddy, 2013). However, due to a large number of CYP450 gene family members, it remains challenging to fully understand their roles in a variety of biosynthetic environments.

The advancement of RNA-seq has made it easier to identify CYP450s and UGTs. Transcriptome analysis of Panax notoginseng (Liu et al., 2015) revealed 350 and 342 predicted unigenes encoding CYP450s and UGTs, respectively; identified 233 CYP450 and 269 UGTs through RNA seq sequencing of Ilex plants, of which 14 CYP450s and one UGTs are considered to play a role in triterpene diosgenin biosynthesis (Wen et al., 2017). In addition, Cheng et al. (2020b) found that CYP716A295 and CYP716A296 to be candidate genes related to oleanolic acid biosynthesis, while CYP72A763 and CYP72A776 are involved in diosgenin biosynthesis in the bark of Aralia elata (Miq.). Dioscorea zingiberensis is diosgenin-rich plant, Li et al. (2018) conducted a comparative transcriptomic analysis of its rhizomes and found that a total of 485 annotated CYP450s unigenes, 195 annotated UGTs unigenes, and 165 CYP unigenes related to the diosgenin biosynthesis. Phylogenetic analysis revealed that four of these CYP candidate genes were most likely involved in the biosynthesis of diosgenin from cholesterol. However, it is unclear whether D. cirrhosa has diosgenin specific to the Dioscoreaceae. To data, no systematic investigation has been conducted to identify the genes related to diosgenin biosynthesis, and the transcriptome database of this plant is unavailable.

Diosgenin, a plant steroid, has demonstrated various pharmacological effects such as anti-hypercholesterolemia, anti-tumor, immune regulation, anti-cancer (Huang, Liu & Jan, 2010; Gong, Qin & Huang, 2011; He et al., 2012; Khan et al., 2022). The biosynthetic pathway of diosgenin in plants has been identified in traditional Chinese medicine plants such as Dioscorea zingiberensis, Dioscorea nipponica, Trigonella foenum-graecum (Li et al., 2018; Sun et al., 2017; Zhou et al., 2019). The biosynthesis process can be divided into several stages. First, acetyl-coenzyme A generates isopentenyl diphosphate (IPP) and dimethylallyl diphosphate (DMAPP) through the MVA or MEP pathway. Next, Farnesyl pyrophosphate is formed from IPP and DMAPP by farnesyl diphosphate synthase (FPS). Squalene synthase (SS) and squalene epoxidase (SE) then convert farnesyl pyrophosphate into 2,3-oxysqualene. This compound is further converted into cycloartenol and lanosterol, respectively, by cycloartenol synthase (CAS) and lanosterol synthase (LSS). In the subsequent stage, multi-step enzymatic reactions transform cycloartenol into cholesterol, which is then oxidized, glycosylated, and cyclized to produce diosgenin. The MVA pathway takes place in the cytoplasm, whereas the MEP pathway occurs in the endoplasmic reticulum (cytosol), mitochondria (or golgi apparatus), and plastids.

While the enzymes involved in the upstream synthesis pathway of diosgenin biosynthesis have been identified, only a few of the downstream synthesis pathways have been characterized, especially the key enzymes that catalyze the formation of diosgenin. For instance, down-regulation of the steroid specific glucosyltransferase TFS3GT2 in fenugreek leads to a reduction in diosgenin levels, suggesting its involvement in diosgenin biosynthesis in fenugreek (Gao et al., 2021); In Polyporus umbellatus, the upregulation of C-8-sterol isomerase, sterol-C-24 methyltransferase (SMT) and sterol-22-desaturase has been found to promoted the biosynthesis of sterol metabolites (Xing et al., 2022), while sterol-C-24 methyltransferase has been shown to catalyze ergosterol biosynthesis (Azam et al., 2014). These downstream enzymes play an important role in the regulation of triterpene biosynthesis in many plants (Phillips et al., 2006; Suzuki & Muranaka, 2007).

Currently, no studies have been conducted on the presence of diosgenin in D. cirrhosa, and the biosynthesis pathway of diosgenin is not well characterized. The lack of molecular-level on this pathway hinders in depth research. Thus, it is of great significance to clarify the biosynthesis pathway and metabolic regulation network of diosgenin in D. cirrhosa tubers, particulary in the context of polygenic family protease, using advanced sequencing and bioinformatics technology. Such investigations not only aid in fundamental research but also facilitate the optimization of the target compound synthesis pathway in the future.

Materials & Methods

Plant material

Four types of tubers, LR (light red), RD (red), DR (dark red), and BR (brownish red), each with distinct colors, were collected from the natural habitat located in the hilly region of Shunde, Guangdong Province, China (112.65°E, 22.88°N). The color gradient of the tubers was compared using color cards, which were easily visible (Fig. 1). Three independent biological replicates were used in this study. The flesh of tubers was quickly frozen in liquid nitrogen and stored at −80 °C for subsequent experiments.

Figure 1 The tuber color differences of D. cirrhosa.

LR indicates light red tuber, RD indicates red tuber, DR indicates dark red tuber, and BR indicate brownish red tuber.

Identification of diosgenin metabolites by UPLC-MS/MS

For sample extraction, the samples were freeze-dried in a vacuum freeze dryer (Scientz-100F), and then ground into powder (30 Hz, 1.5 min) using a mixer mill (MM 400, Retsch); Weigh 100 mg of powder and dissolve in 1.2 ml of 70% methanol solution, vortex for 30 s every 30 min for six times in total, repeat six times, and store in a 4 °C refrigerator overnight. After centrifuge at 12,000 rpm for 10min, then supernatant was collected, filter through a microporous filter membrane (SCAA-104, 0.22 µm pore size), and stored for further UPLC-MS/MS (UPLC, ultra performance liquid chromatography; Shimpack UFLC SHIMADZU CBM30A; MS, Applied Biosystems 4500 Q TRAP) analysis.

The metabolites were identified and quantified by UPLC-MS/MS technology with the following analytical conditions: UPLC: column was an Agilent SB-C18 (1.8 µm, 2.1 mm* 100 mm); Mobile phase: phase A is ultrapure water with 0.1% formic acid, phase B is acetonitrile with 0.1% formic acid; Sample measurements were performed with a gradient program that employed the starting conditions of 95% A, 5% B. Subsequently, a composition of 95% A, 5.0% B was adjusted within 1.10 min and kept for 2.9 min. The flow velocity was set as 0.35 ml per minute; The column oven was set to 40 °C; The injection volume was 4 µl; The column temperature is 40 °C. The effluent was then connected to an ESI-triple quadrupole-linear ion trap (QTRAP)-MS.

The mass spectrometric conditions used for this study involved a QQQ linear ion trap mass spectrometer (Q trap) and API 4500 Q trap LC/MS/MS system to carry out LIT and three quadrupole (QQQ) scanning. The system is equipped with a ESI turbo ion spray interface, runs in positive and negative two ion modes, and is controlled by the QQQ software. ESI source operation parameters are as follows: ion source and turbine spray. Source temperature 550 °C; ion spraying voltage, 5500V (positive ion mode)/-4500V (negative ion mode); The ion source gas I, gas II and curtain gas are set at 55, 60 and 25.0 psi respectively; The collision-induced ionization parameter is set to high. Respectively use 10 and 100 µmol/L polypropylene glycol solutions were respectively used to carry out the instrument tuning and mass calibration. The QQQ scanning used multiple reaction monitoring (MRM) mode, with nitrogen set as the collision gas (nitrogen) at medium. Through further optimization, the corresponding parameters for each MRM ion pair were optimized, and a specific set of MRM ion pairs was monitored based on the metabolites eluted in each period.

Qualitative and quantitative analysis of metabolites

Metabolites quantification is achieved through MRM mode analysis using triple quadrupole mass spectrometry. In this mode, the four-stage rod first screens the precursor ions of the target substance, then eliminates the precursor ions of other molecular weight substances, and finally eliminates the interference of non-target ions, so as to improve the quantitative accuracy and repeatability. After obtaining the mass spectrum data of different metabolites, the mass spectrum peak area is integrated and corrected, and the mass spectrum peak area corresponds to the relative content of substances (Fraga et al., 2010). Analyst (1.6.3) software was used to process mass spectrometry data.

To eliminate the interference of isotopic signals, including repeated signals of K+, Na+, NH4+ ions, and repeated signals of fragment ions that are other substances with larger molecular weight. Metabolite characterization is based on MWDB (metal database). Based on the secondary spectrum information, metabolite data are matched to the corresponding substance information in the database, enabling determination of the metabolites category.

Significance analysis and correlation analysis

The significance of diosgenin metabolite data and gene expression data was analyzed by t-test method with ggsignif package in R. The correlation of metabolites and genes used the corr() function of R software and calculates the correlation coefficient r2 and significance level P value based on Pearson correlation analysis method. A P-value < 0.05 indicates significance, P-value < 0.01 indicates extreme significance. Metabolites or genes with significance level P < 0.05 and —r2— > 0.9 are selected and importing into Cytoscape (3.8.2) software to construct the connection network.

Acquisition, alignment and annotation of transcriptome data

The mRNA library for each sample were constructed and sequenced using the Illumina Hiseq platform by the paired-end reads. The statistical power of this experimental design, calculated in RNASeqPower is 0.91. The raw data of 12 cDNA libraries were uploaded to the NCBI database (accession number: PRJNA741609). Clean reads were obtained by removing low-quality bases and adapter sequences. Trinity software (version 2.6.6) (Grabherr, Haas & Yassour, 2011) was used to assemble clean reads, and the longest clusters obtained by Corset (version 1.07) (Davidson & Oshlack, 2014) were regarded as unigenes. The gene expression level in each sample was calculated using FPKM (Wang, Wang & Li, 2012; Pertea et al., 2015).

Identification of transcription factors (TFs)

The Itak software (1.7A) (Zheng et al., 2016) was used to predicted the transcription factors, which integrates two plant databases: PlnTFDB and PlantTFDB. The identification of transcription factors was carried out using HmmScan comparison through the transcription factor families and rules defined in the database.

Gene expression analysis using qRT-PCR

To verify the expression of diosgenin-related genes, quantitative real-time PCR (qRT-PCR) was performed. Ten diosgenin pathway genes were selected, including IDI gene, DXS gene, MVK gene, AACT gene, SMT gene, SS gene, gcpE gene and ispD gene (Table S7). The NCBI database was used to design primers, and DC18S was used as an internal reference gene, transcribed and amplified using Goldenstar RT6 cDNA Synthesis Mix and T5 fast qPCR mix (SYBR Green I), and three biological repeats were set. qRT-PCR was performed on FQD-96A system (Hangzhou BORI). The procedure is as follows: 95 °C for 2 min, 95 °C cycles 40 times for 15 s, 60 °C cycles for 15 s and 72 °C cycles for 20 s. The relative gene expression was calculated by 2−ΔΔCt method.

CYP450 gene family identification and phylogenetic analysis

The identified CYP450 gene was compared with the CYP450 gene sequence found in this study by using cluster X 2.0.12 (Larkin et al., 2007), and the positions with high vacancy value and missing data percentage were adjusted. The phylogenetic tree was constructed by MEGA 6 software. with the best evolutionary model determined using J model test software. The construction of maximum likelihood (ML) tree is based on TN (Tamura NEI) model (Tamura et al., 2013). Bootstrap is set under 1,000 repetitions to evaluate the importance of nodes.

Results

Identification of diosgenin metabolites

A total of 13 diosgenin metabolites were identified using the HPLC-MS/MS method (Fig. 2, Table S1). Based on the change in metabolite content, they can be divided into three categories (Fig. 3). The first category comprises eight metabolites: 3-O-(2-O-Acetyl-glucosyl) oleanolic acid, protodioscin, trillin-6′- O-sophorotrioside, trillin-6′-O-glucoside, diosgenin-3-O-glucoside (Trillin), pseudogenin B (parisyunnanoside b), diosgenin-3-O-glucosyl (1 →4) rhamnosyl (1 →4) rhamnosyl (1 →2) glucoside (diosgenin-3-O-glcosyl (1 →4) rhamnosyl(1 →4) rhamnosyl (1 →2) glcoside) and pseudoprotodioscin. The highest content of these metabolites was observed in DR (Dark red) followed by LR (Light red), RD (Red) and BR(Brownish red). The second category includes three diosgenin metabolites: diosgenin-3-O-rhamnosyl - (1 →3) glcoside, diosgenin-3-O-rhamnosyl (1,2) glcoside, ruscogenin-1-O-xylosyl (1,3) fucoside. Only BR exhibited high content while the LR, RD, and DR showed low content. The third category includes two diosgenin metabolites, including gracillin and pennogenin-3-O-glucoside, and RD and DR tubers showed relatively high content. Therefore, diosgenin metabolites are not accumulated gradually in D. cirrhosa tubers at all color stages. Based on the results above, diosgenin metabolites primarily accumulate in DR tubers, followed by BR tubers.

Identification of diosgenin biosynthesis genes

In our study, we identified 43 unigenes encoding 21 key enzymes involved in diosgenin biosynthesis, which includes two metabolic pathways: IPP and MEP (Table 1). Among these, seven unigenes were identified in the MEP pathway, including one DXS gene, one DXR gene, five ISP genes, one gcpE gene and three FPS genes. In MVA pathway, we identified four AACT genes, one HMGS gene, eight HMGR genes, three MVK genes, one PMK gene, one MVD gene and two IDI genes (Table 2, Table S2).

Figure 2 Differences of diosgenin metabolites in D. cirrhosa tubers.

Three biological replicates are shown in boxplot. LR: light red; RD: red; DR: dark red; BR: brownish red. Each color represents one tuber, an asterisk (*) indicates significance level and NS indicates non-significant (* p < 0.05, ** p < 0.01, *** p < 0.001).

Figure 3 Diosgenin related metabolites in D. cirrhosa tubers.

The metabolites were scaled using Z-score of relative content (mean value of three replications) in the heatmap. LR, light red; RD, red; DR, dark red; BR, brownish red.

Table 1 Genes involved in diosgenin biosynthesis pathway in D. cirrhosa.

Gene	Enzyme	Number	KEGG	
SQE	Squalene epoxidase	1	K00511	
FPS	Farnesyl pyrophasphate synthase	3	K13789	
C14-R	Sterol C14-reductase	2	K00222	
C5(6)	Sterol desaturase	2	K00227	
MVK	Mevalonate kinase	3	K00869	
HMGCR	Hydroxymethylglutaryl-CoA reductase	8	K01661	
DXS	1-deoxy-D-xylulose-5-phosphate synthase	1	K01662	
ispE	4-diphosphocytidyl-5-C-methyl-D-erythritol kinase	2	K00919	
ispF	2-C-methyl-D-erythritol 2,4-cyclodiphosphate synthase	1	K01770	
gcpE	(E)-4-hydroxy-3-methylbut-3-enyl-diphosphate synthase	1	K03526	
ispH	4-hydroxy-3-methylbut-2-en-4-yl diphosphate reductase	1	K03527	
IDI	Isopentenyl-diphosphate Delta-isomerase	2	K01823	
DXR	1-deoxy-D-xylulose-6-phosphate reductoisomerase	1	K00099	
HMGCS	Hydroxymethylglutaryl-CoA synthase	1	K01641	
CAS	Cycloartenol synthase	1	K01853	
ispD	2-C-methyl-D-erythritol 11-phosphate cytidylyltransferase	1	K00991	
SS	Squalene synthase	3	K00919	
PMK	Phosphomevalonate kinase	1	K00938	
MVD	Diphosphomevalonate decarboxylase	1	K01597	
AACT	Acetyl-CoA C-acetyltransferase	4	K00626	
SMT	Sterol 24-C-methyltransferase	3	K00559	
	All	43	24	

Table 2 TFs identified in D. cirrhosa.

TFs	Number	LR vs RD (All/up)	RD vs DR (All/up)	DR vs BR (All/up)	
AP2/ERF	82	26/9	56/47	50/15	
MYB	88	42/16	54/30	48/29	
WRKY	50	23/9	39/26	25/9	
NAC	51	26/11	33/25	27/8	
C2H2	56	27/14	31/16	34/23	
bZIP	55	30/14	29/15	31/21	
bHLH	53	24/14	28/13	22/14	
C3H	52	16/11	24/7	34/27	
SET	32	14/12	9/1	17/16	
HB-HD-ZIP	31	16/7	22/15	18/12	
GRAS	23	15/7	19/12	13/4	
C2C2-Dof		14/5	15/10	13/10	
SNF2	49	23/22	15/1	31/31	
PHD	31	11/9	14/4	19/18	
GARP-G2-like	23	13/8	13/5	21/16	
AUX/IAA	20	7/3	12/10	13/6	

The oxidized squalene cyclase (OSC) family, comprising cycloartenol synthase (CAS), lanosterol synthase (LSS) and amyrin synthase (AS), cyclize 2,3-oxide squalene to synthesize different phytosterol skeletons. OSC enzymes are the key nodes in the triterpene biosynthesis pathway, and different members of the enzyme family have been shown to produce various triterpene skeletons in Arabidopsis and rice (Phillips et al., 2006; Morlacchi et al., 2009). Ohyama et al. (2009) demonstrated that phytosterols are synthesized via a dual pathway of cycloartenol and lanosterol, which are precursors of sterols and steroid hormones, while the genes encoding wood sterol biosynthesis have only been identified in a few plant species such as Arabidopsis, rice, Panax ginseng and Dioscorea zingiberensis. We found only one unigene (cluster-6992.57030) with high homology with the CAS and no LSS gene in this study. In addition, two C5 (6) genes, three C14-R genes and three unigenes encoding sterol 24-C methyltransferase (SMT) were identified, which are key genes downstream of diosgenin biosynthesis pathway and may play a role in the diosgenin conversion.

Potential biosynthesis pathway of diosgenin and expression of related genes

The expression level (FPKM) of diosgenin genes and the potential pathway of D. cirrhosa diosgenin biosynthesis are presented in Fig. 3. It was showed that the MVA and MEP pathways exhibited a “high-low-high-low” pattern, except for the ispH gene. Most genes in the diosgenin biosynthesis pathway, such as cluster-6992.39943 (DXS), cluster-6992.29239 (DXR), cluster-6992.34108 (ispD), cluster-6992.46302(ispE) and cluster-6992.52592 (ispF), were highly expressed in LR and DR but weakly expression in RD. This expression pattern was consistent with the change pattern of the first of diosgenin metabolites (Fig. 2). It was worth noting that the expression levels of cluster-6992.49865 (HMGR), cluster-6992.55638 (FPS) and cluster-6992.39611 (SE) were significantly upregulated in BR compared with the other three groups, which is consistent with the second category of diosgenin metabolites (Fig. 3). These results indicate that the genes exhibiting expression patterns consistent with the change levels of diosgenin metabolites may be the be the key genes controlling diosgenin biosynthesis in D. cirrhosa tubers.

Moreover, in the downstream of the diosgenin biosynthesis, we identified three SMT genes, one CAS gene, two C14-R genes and two C5(6) genes. Among them, three SMT genes (cluster-6992.56623, cluster-6992.57489 and cluster-6992.31684) exhibited the same expression pattern as cluster-6992.57030 (CAS), cluster-6992.40736 (C5(6)) and cluster-6992.50375 (C14-R) (Fig. 4). It is worth noting that enzymes such as C-16, C-22, and C-26 are crucial in the conversion of cholesterol to diosgenin, but they were not identified in this study (Zhou et al., 2019). Therefore, we propose that diosgenin in D. cirrhosa may be a precursor of diosgenin catalyzed by cycloartenol via the SMT enzyme and that diosgenin is subsequently further synthesized by sterol desaturase (C5 (6)) and sterol C14 reductase (C14-R). These finding suggest that the genes downstream of the diosgenin biosynthesis pathway may also be key genes controlling diosgenin biosynthesis in D. cirrhosa tubers.

Figure 4 Diosgenin biosynthesis pathway and related genes expression in D. cirrhosa.

AACT, Acetoacetyl-CoA thiolase; HMGS, hydroxy-3-methylglutaryl-CoA Synthase; HMGR, 3-hydroxy-3-methylglutaryl-CoA Reductase; MVK, Mevalonate kinase; PMK, 5-phosphomevalonate kinase; MVD, Diphosphomevalonate decarboxylase; DXS, 1-deoxy-D-xylulose-5-phosphate synthase; DXR, 1-deoxy-D-xylulose-5-phosphate reductoisomerase; ispD, 2-C-methyl-D-erythritol 11-phosphate cytidylyltransferase; ispE, 4-diphosphocytidyl-5-C-methyl-D-erythritol kinase; ispF, 2-C-methyl-D-erythritol 2,4-cyclodiphosphate synthase; gcpE, (E)-4-hydroxy-3-methylbut-3-enyl-diphosphate synthase; ispH, 4-hydroxy-3-methylbut-2-en-4-yl diphosphate reductase; FPS, farnesyl pyrophasphate synthase; SS, squalene synthase; SE, squalene epoxidase; CAS, cycloartenol synthase; LSS, lanosterol synthase; SMT, sterol 24-C-methyltransferase; C5(6), sterol desaturase; C14-R, sterol C14-reductase; UGT, uridine diphosphate (UDP)-dependent glycosyltransferases. The color scale represents the average of FPKM value (scaled using Z-score), red color indicates high expression and purple color indicate low expression.

Transcription factor (TF) identification

To identified TFs, we searched the plant TF databases and identified 3365 unigenes belonging to 82 TF families in the D. cirrhosa transcriptome dataset. Differential expression analysis revealed that 1261 TFs were differentially expressed, with 88 MYB family members identified, including 5 from the R2R3-MYB subfamily. Additionally, 55, 53, 50, 51, 56, 52, 49, and 20 genes belonged to the bZIP, bHLH, WRKY, NAC, C2H2, C3H, SNF2, and Aux/IAA families, respectively (Table 3). These diverse TF families provide valuable information for further biological analysis.

Table 3 Information of published CYP450 genes.

Gene	Accession No.	Species	
Gm CYP74A1	NM_001289366.1	Glycine max	
Sl CYP51	NM_001247608.2	Solanum lycopersicum	
Sc CYP51	AY552551.1	Solanum chacoense	
Sb CYP51	U74319.1	Sorghum bicolor	
Sl CYP734A7	NM_001247011.2	Solanum lycopersicum	
Pk CYP73A	D82812.1	Populus kitakamiensis	
Pg CYP716A53v2	JX036031.1	Panax ginseng	
Mt CYP93E2	DQ335790.1	Medicago truncatula	
Mt CYP716A12	DQ335781.1	Medicago truncatula	
Ms CYP716A12	KM978958.1	Medicago sativa	
At CYP724A1	NM_121444.4	Arabidopsis thaliana	
GS CYP75A	MW298105.1	Glycine soja	
Gm CYP93E1	NM_001249225.2	Glycine max	
Mt CYP93E2	DQ335790.1	Medicago truncatula	
Gu CYP93E3	AB437320.1	Glycyrrhiza uralensis	
At CYP734A1	NM_128228.4	Arabidopsis thaliana	
Gu CYP88D6	AB433179.1	Glycyrrhiza uralensis	
At CYP86A1	NM_125276.3	Arabidopsis thaliana	
Pg CYP716A47	JN604537.1	Panax ginseng	
At CYP90B1	AF044216.1	Arabidopsis thaliana	
At CYP86A8	AJ301678.1	Arabidopsis thaliana	
At CYP84A1	AY666123.1	Arabidopsis thaliana	
As CYP51H10	DQ680849.1	Avena strigosa	
At CYP51	AY666123.1	Arabidopsis thaliana	

Furthermore, TFs with FPKM value >1 in at least one sample were selected, resulting in 37 screened genes, including MYBs, bHLHs, WRKYs, and plant hormone related TFs (Table S3). We analyzed their expression patterns and depicted their expression level in heatmap (Fig. S1). the DR tuber had the largest number of highly expressed TFs, including Aux/IAAs, WRKYs and AP2s, followed by BR and RD tubers, while LR tubers had the lowest expression levels. These results indicate that the expression of TFs was highest in the middle and late stage of tuber color formation and lowest in the early stage of color formation.

Interaction between diosgenin biosynthesis gene and TFs

In order to better understand the relationship between diosgenin biosynthesis genes and TFs, we calculated the Pearson correlation coefficient (r2) between TFs and diosgenin biosynthesis genes. We then selected the items with —r2— > 0.9 and used them to construct a gene interaction network. The results showed that a total of 22 TFs were strongly associated with diosgenin genes (—r2— > 0.9, P < 0.05), including WRKY, MYB, bZIP, bHLH, Aux/IAA and AP2/ERF gene family. Among them, bZIP had the most connections with diosgenin biosynthesis genes, followed by MYB and AP2/ERF (Fig. 5, Table S4). These candidate TFs are likely to play a crucial role in the biosynthesis of diosgenin in D. cirrhosa.

Figure 5 Regulatory networks of TFs and diosgenin genes.

Each yellow circle represents a TF, and each blue circle represents diosgenin a gene.

CYP450 gene family analysis and phylogenetic tree construction

The expression patterns of genes related to secondary metabolite generally correspond metabolites levels in different parts of plant. CYP450 family is the largest plant protein family and plays important roles in catalyzing most of the oxidative steps in plant secondary metabolism. It is well known that CYP450 is involved in the catalytic reaction of cholesterol formation from lanosterol in the diosgenin biosynthesis pathway (Koolman & Roehm, 2005). However, few CYP450 or UGT genes involved in diosgenin biosynthesis have been identified in Dioscorea zingiberensis and fenugreek (Zhou et al., 2021; Cheng et al., 2020a). In this study, 206 unigenes encoding CYP450 protein were identified in D. cirrhosa transcriptome data through different database annotations (Fig. 6). Among them, 112 belong to CYP2 subfamily, 68 belong to CYP4/CYP19/CYP26 family and four belong to CYP3/CYP5/CYP6/CYP9 family. A total of 40 candidate genes were obtained by screening unigene with FPKM >1 in at least one tissue, which are considered potential genes for diosgenin biosynthesis in D. cirrhosa. Among these candidate genes, 14 genes were identified as members of the CYP71A1 subfamily, five genes were identified as members of the CYP72A219 subfamily, four genes identified as members of the CYP94C1 subfamily and three genes identified as members of the CYP711A1 subfamily. Through homologous annotation analysis, we found that 12 genes were homologous to Phoenix dactylifera, 20 genes were homologous to Elaeis guineensis, four genes were homologous to Musa acuminata subsp. Malacensis, one gene was homologous to Daucus Carota subsp. Sativus, one gene was homologous to Ananas comosus, and one gene was homologous to Asparagus officinalis (Table S5).

Figure 6 The expression patterns of CYP450s in D. cirrhosa tubers.

Z-score obtained from averaged FPKM of three biological replicates was used. Red color indicates high expression and yellow color indicate low expression.

Moreover, among the CYP450 genes identified, the genes cluster-6992.48174 and cluster-6992.33292 were annotated as steroid-22- α-hydroxylase (CYP90B), and the gene cluster-6992.33753 and cluster-6992.33754 were annotated as sterol-14-demethylase (CYP51) (Fig. 7, Table S5). Both steroid-22-α-hydroxylase and sterol-14-demethylase are downstream synthase that catalyze diosgenin biosynthesis, indicating that these CYP450 genes may be involved in diosgenin synthesis in D. cirrhosa tubers. Further analysis of the expression level of these genes and search of homologous annotation database showed that six CYP450 genes were highly expressed in LR, 13 CYP450 genes were highly expressed in RD and 22 CYP450 genes showed high expression level in DR. It should be noted that the expression levels of almost all CYP450 genes were low during the BR period.

Figure 7 Expression levels of CYP450 genes between D. cirrhosa tubers.

The genes annotated as steroid-22- α-hydroxylase (CYP90B) and sterol-14-demethylase were marked by the blue arrow. Z-score obtained from averaged FPKM of three biological replicates was used. Red color indicates high expression and blue color indicate low expression.

Diosgenin is synthesized from cholesterol through a series of oxidation reactions at C-22, C-26 and C-16 positions (Zhou et al., 2019). Steroid-22- α-hydroxylase and sterol-14-demethylase play a catalytic role in the sequential stages of cholesterol formation, which is the precursor of diosgenin biosynthesis (Cao et al., 2021). In addition, the identification of unigenes associated with sterol-14-demethylase and steroid-22- α-hydroxylase in our CYP450 candidate list suggests that some other CYP candidate genes may play important roles in diosgenin biosynthesis. To understand the functions of these CYP candidate genes, we performed phylogenetic analysis on the CYP450 candidate genes along with other well-characterized CYP genes from various metabolic pathways, including those involved in the biosynthesis of triterpenes, diosgenin, and flavonoid (Table 4).

In Fig. 8, unigene cluster-6992.63413 clustered with clusters-6992.48174 and cluster-6992.33292 from the CYP90B subfamily and AtCYP90B1 from Arabidopsis (Fujita et al., 2006), which is known to have sterol C-22 hydroxylase in the brassinosteroid pathway. This suggests that unigene cluster-6992.63413 may be a CYP450 candidate responsible for steroid C-22 hydroxylation in diosgenin biosynthesis. Unigene cluster-6992.33753 and cluster-6992.33754 are clustered into the same branch with AtCYP51 of Arabidopsis (Kim et al., 2005), SiCYP51 of tomato(Solanum lycopersicum) and SbCYP51 of Sorghum bicolor, which are characterized by sterol-14-demethylase. Cluster-6992.51327 is more closely related to SbCYP51 of Sorghum bicolor, suggesting that it may encode sterol-14-demethylase in diosgenin biosynthesis pathway. Unigene cluster-6992.67001, cluster-6992.40273 and β-amyrin C-24 oxidases are relatively closer, including MtCYP93E2 from tomato (Li et al., 2007), GuCYP93E3 from Glycyrrhiza uralensis (Seki et al., 2008), and GmCYP93E1 from potato (Shibuya et al., 2006), indicating that unigene cluster-6992.67001 and cluster-6992.40273 may encode amyrin oxidase in D. cirrhosa diosgenin biosynthesis. Notably, AtCYP734A1 (Lin et al., 1999) in Arabidopsis and CYP enzyme SlCYP734A7 in tomato are characterized by sterol C-26 hydroxylase in the biosynthesis of diosgenin (Vasav & Barvkar, 2019), which are far away from each other in branches. In addition, unigene cluster-6992.77886, cluster-6992.66001, cluster-6992.54954 and cluster-6992.41463 clustered into the same branch as steroid C-26 hydroxylase gene AtCYP734A1 (Lin et al., 1999) of Arabidopsis thaliana, β-amyrin C-11 oxidase gene of Medicago truncatula, MtCYP716A12 (Li et al., 2007), gibberellin biosynthesis gene of Cucurbita maxima, and GmCYP88A (Helliwell et al., 2001). This suggests that these genes may be multifunctional synthases, and their functions require further investigated by cloning full-length sequences.

Table 4 The FPKM value of UTGs in D. cirrhosa.

Gene ID	LR	RD	DR	BR	Predited	
Cluster-6992.52789	7.71	10.63	24.81	2.76	Elaeis guineensis	
Cluster-6992.47459	35.42	2217.30	33.54	360.04	Elaeis guineensis	
Cluster-6992.41090	1877.13	2848.44	2194.17	250.56	Phoenix dactylifera	
Cluster-6992.37602	2879.41	1208.69	1205.58	2108.93	Phoenix dactylifera	
Cluster-6992.31856	11.11	8.85	20.45	7.34	Elaeis guineensis	

Figure 8 Phylogenetic analysis of CYP450 candidates from D. cirrhosa transcriptome and previously published CYP450s from various metabolic pathways.

The genes with red dots indicate hub genes associated with sterol C-22 hydroxylase and sterol-14-demethylase.

Further correlation analysis between the CYP450 candidate genes and diosgenin metabolites showed that 14 CYP450 genes had high correlation with the content of seven diosgenin metabolites (—r2— > 0.9, Table S6). These CYP450 unigenes showed a consistent accumulation pattern with diosgenin metabolites and are believed to be key genes in diosgenin biosynthesis.

Glycosylation has a crucial role in the biological activity of diosgenin compounds, and diphosphate (UDP)-dependent glycosyltransferases catalyze a crucial step in the biosynthesis of diosgenins (Sawai & Saito, 2011). To provide energy for metabolite biosynthesis, sugar transformation is necessary, and we identified five UGTs in the transcriptome data of D. cirrhosa. By performing a gene homology search in the NR database, we found that these genes have homology with sucrose biosynthesis in Elaeis guineensis and Phoenix dactylifera (Table 5).

Table 5 The FPKM value of UTGs in D. cirrhosa.

Gene ID	LR	RD	DR	BR	Predited	
Cluster-6992.52789	7.71	10.63	24.81	2.76	Elaeis guineensis	
Cluster-6992.47459	35.42	2217.30	33.54	360.04	Elaeis guineensis	
Cluster-6992.41090	1877.13	2848.44	2194.17	250.56	Phoenix dactylifera	
Cluster-6992.37602	2879.41	1208.69	1205.58	2108.93	Phoenix dactylifera	
Cluster-6992.31856	11.11	8.85	20.45	7.34	Elaeis guineensis	

Identification of key genes related to diosgenin biosynthesis

We utilized Pearson correlation analysis to explore the relationship between the genes related to diosgenin biosynthesis and metabolites. The results indicated a strong correlation between 22 diosgenin synthesis related genes and 12 diosgenin metabolites (p < 0.05, —r2— > 0.95). Among these genes, 12 genes showed a significantly positively correlated with metabolites, including two UDPGs genes, one DXS gene, one gcpE gene, two SMT genes, two IDI genes, one SS gene, one MVK gene, one AACT gene and one ispD gene; 10 genes were negatively correlated with metabolites, including one DXR gene, one C5(6) gene, two HMGCR genes, one CAS gene, one AACT gene, one HMGS gene, one FDS gene, one UDPGs gene and one SE gene (Fig. 9). These genes are speculated to play either positive or negative regulatory roles in the diosgenin biosynthesis.

Figure 9 Heatmap of Pearson correlation analysis between diosgenin genes and diosgenin metabolites.

The value in the box represents the r2 value, and 0.00 indicates no significance. D.1, ruscogenin-1-O-xylosyl(1,3) fucoside; D.2, diosgenin-3-O-rhamnosyl(1,2) glcoside; D.3, Diosgenin-3-O-rhamnosyl-(1 →3) glcoside; D.4, gracillin; D.5, pennogenin-3-O-glucoside; D.6, pseudoprotodioscin; D.7, diosgenin-3-O-glcosyl (1 →4) rhamnosyl (1 →4) rhamnosyl (1 →2) glcoside; D.8, trillin (diosgenin-3-O-glucoside); D.9, trillin-6′-O-glucoside; D.10, parisyunnanoside B; D.11, trillin-6′-O-sophorotrioside; D.12, 3-O-(2-O-acetyl-glucosyl) oleanolic acid.

Quantitative Real-Time (qRT-PCR) validation of diosgenin genes

To validate the accuracy of our RNA-seq data, we conducted qRT-PCR to evaluate the gene expression of 10 diosgenin genes at the transcriptional level (Fig. 10). The results were consistent with transcriptome data, suggesting the reliability of our findings.

Figure 10 RT-PCR of key genes in diosgenin biosynthesis pathway.

Results represent mean values ± SE from three biological replicates. Asterisks indicates the statistical significance of the difference between LR and RD, RD and DR, DR and BR, r-value above 0.7 indicates a very close relationship (***), a range of 0.4 to 0.7 indicates a close relationship (**), and a range of 0.2 to 0.4 indicates a general relationship (*), “NS” indicates not significant.

Discussion

Diosgenin, a triterpene-derived steroid, is involved in the biosynthesis of isoprenoids, steroids, sesquiterpenes and triterpenes in higher plants. The key enzyme genes in the terpene biosynthesis pathway have been proved to play an vital role in the regulating terpene biosynthesis. For example, up-regulating the DXS and DXR genes in the hairy roots of sage promotes the accumulation of diterpenes  (Vaccaro et al., 2019); PGSQE1 promotes ginsenoside biosynthesis in ginseng by regulating its expression (Han et al., 2010). In Acanthopanax senticosus, the expression of β-amyrin synthase ESbas promotes the accumulation of oleanolic acid (Jo et al., 2017). Diosgenin can be synthesized in higher plants via the MVA or MEP pathways, with MVA being the preferred pathway (Nes, 2011). In this study, we focused on the diosgenin biosynthesis pathway in D. cirrhosa, and proposed that this synthesis involving both MEP and MVA pathway. A total of 43 unigenes encoding up to 21 key enzymes involved in diosgenin biosynthesis. The expression pattern of most genes were consistent with that of diosgenin metabolites, which were the key genes to control diosgenin biosynthesis. Interestingly ,this study found that the expression pattern of D. cirrhosa diosgenin biosynthesis genes did not increase gradually with the accumulation of tuber pigment. This identification provides candidate genes for further gene manipulation.

In recent years, there have been studies investigate the regulation of diosgenin metabolites biosynthesis by plant hormones. For example, Sun et al. (2017) found that the rhizome of D. cirrhosa treated with methyl jasmonate promote the biosynthesis of diosgenin and the significant expression of diosgenin genes, but inhibited the synthesis of C-24 methylation product. In addition, ethylene treatment was found to enhance the accumulation of diosgenin by upregulating the expression of HMGR and CAS genes in D. zingiberensis (Diarra et al., 2013), while hormone treatments such as salicylic acid and methyl jasmonate can promote the expression of DXD gene in Ginkgo biloba roots, stems, leaves, pericarps and seeds (Gong et al., 2006). A novel discovery of this study is the significant strong connection between hormone related transcription factors and diosgenin biosynthesis genes (—r2— > 0.9, P < 0.05). Hence, we propose that diosgenin biosynthesis may also be regulated by TFs such as bZIP, MYB and AP2/ERF, and diosgenin pathway genes may function together with TFs on D. cirrhosa diosgenin biosynthesis.

Recently, a new study demonstrated that the down-expression of TFSMT1 gene in fenugreek resulted in an increase in cholesterol levels, but a significant decrease in diosgenin. Although overexpression of this gene led to an increase in diosgenin content, there was no significant effect on cholesterol biosynthesis. This study suggested that cholesterol does not participate in diosgenin biosynthesis and that SMT gene was closely related to the biosynthesis of diosgenin in fenugreek (Cao et al., 2021). In this study, we identified three SMT genes were expressed in diosgenin biosynthesis, and no cholesterol related metabolites were detected, indicating that D. cirrhosa diosgenin biosynthesis may not go through the cholesterol pathway. This finding suggests that diosgenin biosynthesis in D. cirrhosa may not involve the cholesterol pathway, but instead is synthesized from sitosterol under the catalysis of the SMT gene, which supports the findings of Cao et al. (2021). However, further verification is needed to determine if the downstream synthesis pathway of diosgenin in D. cirrhosa is regulated by the enzyme.

Cytochrome P450 plays a crucial role in the conversion of cholesterol to diosgenin (Christ et al., 2019). To date, 80 CYP450 genes have been identified that are associated with terpene metabolism. The CYP51G, CYP85A, CYP90B-D, CYP710A, CYP724B and CYP734A subfamily members are generally conserved in the plant kingdom and are involved in primary metabolism related to the biosynthesis of plant essential sterols and steroid hormones (Ghosh, 2017). In addition, specialized triterpenoids require the participation of subfamilies, such as CYP51H, CYP71A, D, CYP72A, CYP81Q, CYP87D, CYP88D, CYP93E, CYP705A, CYP708A and CYP716A, C, E, S, U and Y, and the members of these subfamilies may have species-specific functions, including chemical defense against specialized pathogens (Ghosh, 2017). In most cases, cyclic triterpene scaffolds catalyze a large number of scaffold, regional and stereospecific oxidative modifications catalyzed by cytochrome P450 monooxygenase, resulting in triterpene scaffolds with various functional groups, such as hydroxyl, carbonyl and carboxyl. Furthermore, the addition of oxygen function mediated by CYP450s exposes the triterpene scaffold subsequently exposed to UDP glycosyltransferases and acyltransferases (ATS), leading to the formation of diosgenin and acylated triterpenes (Osbourn, Goss & Field, 2011). In recent years, significant progress has been made in understanding the biochemical function of CYP450 involved in plant triterpene metabolism. Combining the genetic screening of mutants with impaired triterpene biosynthesis and the availability of integrated genomic and transcriptome resources, some CYP450s involved in the structural modification of plant triterpenes has been identified (Zeng et al., 2018; Lertphadungkit et al., 2021). In addition, recent advances in transcriptome have contributed to the identification of a large number of CYP450s. Assigning biochemical functions to these CYP450s will aid in studying the biosynthesis of diosgenin in plants.

In this study, a total of 206 unigenes encoding CYP450 proteins were identified, with 112 belong to CYP2 subfamily, 68 belong to CYP4/CYP19/CYP26 family and 4 belong to CYP3/CYP5/CYP6/CYP9 family. Further analysis of the 40 candidate P450 genes identified14 genes identified as CYP71A1 gene, five genes identified as CYP72A219 gene, four genes identified as CYP94C1 gene and three genes identified as CYP711A1 gene. The gene homology annotation analysis mainly divided these CYP genes into species such as Phoenix dactylifera, Elaeis guineensis, Musa acuminata subsp. malaccensis and Ananas comosus. Four key CYP450 genes of diosgenin biosynthase were identified, cluster-6992.48174 and cluster-6992.33292 encode steroid-22- α-hydroxylase, cluster-6992.33753 and cluster-6992.33754 encode sterol-14-demethylase (CYP51). The CYP450 phylogenetic analysis suggested that the gene cluster-6992.63413 may be involved in sterol C-22 oxidation in diosgenin biosynthesis, cluster-6992.51327 may encode sterol-14-demethylase in diosgenin biosynthesis pathway, and cluster-6992.67001 and cluster-6992.40273 may encode amyrin oxidase in diosgenin biosynthesis (β-amyrin-C-24 oxidases). This study systematically analyzed the CYP450 and UGT gene families of D. cirrhosa, laying a foundation for further studies on the functions of these two multi-gene families.

Supplemental Information

Figure S1 The expression level of TFs identified in D. cirrhosa.

The color scale represents the average of FPKM value (scaled using Z-score), red color indicate high expression and blue color indicate low expression.

Click here for additional data file.

Table S1 Diosgenins and their relative contents in D. cirrhosa tubers

Relative content of dioseginin metabolites. HPLC-MS/MS analyses were performed on three biological replicates.

Click here for additional data file.

Table S2 Diosgenin metabolites and their relative contents in D. cirrhosa tubers

Click here for additional data file.

Table S3 Expression level of candidate TFs (FPKM value)

Click here for additional data file.

Table S4 Correlation analysis between TFs and diosgenin genes

Click here for additional data file.

Table S5 Annotation information of candidate CYP450 genes

Click here for additional data file.

Table S6 Correlation analysis between CYP450 gene and diosgenin metabolites

Click here for additional data file.

Table S7 The quantitative real-time (qRT-PCR) primer sequence

Click here for additional data file.

Supplemental Information 9 The checklist of quantitative real-time PCR experiments

Click here for additional data file.

Additional Information and Declarations

Competing Interests

Author Contributions

Data Availability

The authors declare there are no competing interests.

Yan Lin conceived and designed the experiments, authored or reviewed drafts of the article, and approved the final draft.

Qiuyan Hu performed the experiments, prepared figures and/or tables, and approved the final draft.

Qiang Ye analyzed the data, prepared figures and/or tables, and approved the final draft.

Haohua Zhang analyzed the data, prepared figures and/or tables, and approved the final draft.

Ziyu Bao analyzed the data, prepared figures and/or tables, and approved the final draft.

Yongping Li conceived and designed the experiments, authored or reviewed drafts of the article, supervision, and approved the final draft.

Luo Jian Mo conceived and designed the experiments, authored or reviewed drafts of the article, supervision, and approved the final draft.

The following information was supplied regarding data availability:

The sequence reads are available at GenBank: PRJNA741609.

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
