# Peer review of "Diosgenin biosynthesis pathway and its regulation in Dioscorea cirrhosa L"

_PeerJ, doi:10.7717/peerj.16702_

## Round 0.1 · original submission · Minor Revisions

I agree with the reviewers and the article can be accepted after minor revisions.

**Language Note:** The review process has identified that the English language must be improved. PeerJ can provide language editing services - please contact us at copyediting@peerj.com for pricing (be sure to provide your manuscript number and title). Alternatively, you should make your own arrangements to improve the language quality and provide details in your response letter. – PeerJ Staff

Reviewer 1 ·

Basic reporting

Authors conducted HPLC-MS/MS analysis to identify 13 diosgenin metabolites in D. cirrhosa tuber, and utilized transcriptome data to identify 21 key enzymes and 43 unigenes. The results reveal 22 TFs are strongly associated with diosgenin biosynthesis genes. The work is important to aim improving the biosynthesis of diosgenin compounds in plants.

Experimental design

Authors conducted HPLC-MS/MS analysis to identify 13 diosgenin metabolites in D. cirrhosa tuber, and utilized transcriptome data to identify 21 key enzymes and 43 unigenes. Based on the gene family analysis to identified the CYP450 gene family and TF, and to verify the expression of diosgenin-related genes. The work included the bioinformatic analysis and experimental verify.

Validity of the findings

Authors identified 206 CYP450 genes including 40 being potential CYP450s. these CYP450s were associated with sterol C-22 hydroxylase, sterol-14-demethylase and amyrin oxidase in diosgenin biosynthesis. There some comments as below:
1. Figure 10 should check, for the remake of the "*", what is the NS?
2. the tools should be cite, as what software to get the heatmap
3. I sugget authors to compare the expression of hub gene in different population to verify the Dao-Di Herb.

Reviewer 2 ·

Basic reporting

The manuscript “Diosgenin biosynthesis pathway and its regulation in Dioscorea cirrhosa L.” by Yan Lin analyzes the diosgenin synthesis pathway through transcriptome and metabolism data. 13 diosgenin metabolites, 21 key enzymes, and 43 unigenes were identified, which will apply the guidance for future research aimed at improving the diosgenin biosynthesis ability in plants. The manuscript includes a large amount of data and analysis, but it still needs more evidence to support the results and do a thorough investigation of the diosgenin biosynthesis.

1. In line 280, the author said that no “no CYP450 or UGT genes involved in diosgenin biosynthesis have been identified in Dioscoreae or other diosgenin producing plants”, which is not accurate. There are some papers that identified the CYP450 genes involved in diosgenin biosynthesis, such as, “Zhou, Chen, et al. The Plant Journal 109.4 (2022): 940-951.” and “Cheng J., Chen J., Liu X., et al. Plant Communications. 2020. 2(1):100079.” The author should reinvestigate the related paper carefully and correct your manuscript.
2. As there are many studies on other diosgenin-producing plants, the author should make a comparison between your results and previous results and discuss the similarities and differences.

Experimental design

1. The author introduced and compared several important diosgenin metabolites in the manuscript, some of which are not easy to identify. Can the author apply the HPLC/MS chromograph of these compounds?

Validity of the findings

The finding is important.

·

Basic reporting

in all text some times you do not need use some extra words like here PATHWAY
AND There are many examples in the text should be improved
Please see attached file with highlighted to be consider

Experimental design

In gene expression analysis It was written:
The significance of diosgenin metabolite data and gene expression data was analyzed by t-
150 test method with ggsignif package in R.
T-test compares between two group while they have replicates and may have done Analysis of variance

please MAKE CLEAR AND UNDERSTANDABLE THESE SECTION
2.7. Gene Expression Analysis Using qRT-PCR

what kind of primer used for cDNA synthesis?
Is there any expression analysis for leaf tissue?

Validity of the findings

no comment

Additional comments

In paragraph starting line 378:
In this paragraph you should speak about transcription factors rather than hormones since you haven't treat plants with hormone
there fore you don't have data to discuss here
please give reference to this:
Combining the
418 genetic screening of mutants with impaired triterpene biosynthesis and the availability of
419 integrated genomic and transcriptome resources, some CYP450s involved in the structural
420 modification of plant triterpenes has been identified.

---

## Round 0.2 · accepted · Accept

The authors have made changes according to the comments and the current form of the article can be accepted for publication.

Reviewer 1 ·

Basic reporting

there is a minor error, I dont know why authors to get the absolute of R2? R2 all > 0, why? please check this.

Experimental design

no comment

Validity of the findings

no comment

Additional comments

no comment

Reviewer 2 ·

Basic reporting

The author modified the manuscript based on review's suggestion

Experimental design

Although there are some results, such as the HPLC/MS chromograph of some compounds, still not solid enough, but the author make an explanation fot them.

Validity of the findings

a novel and interseting finding

Additional comments

no